# Genome-Wide DNA Methylation Differences between *Bos indicus* and *Bos taurus*

**DOI:** 10.3390/ani13020203

**Published:** 2023-01-05

**Authors:** Xiaona Chen, Xinyu Duan, Qingqing Chong, Chunqing Li, Heng Xiao, Shanyuan Chen

**Affiliations:** 1School of Ecology and Environmental Science, Yunnan University, Kunming 650500, China; 2School of Life Sciences, Yunnan University, Kunming 650500, China

**Keywords:** Yunnan zebu, Holstein cattle, WGBS, DNA methylation, DMR, DMG

## Abstract

**Simple Summary:**

Domestic cattle are one of the indispensable economic animals in human production and life, and zebu (*Bos indicus*), as one type of the domestic cattle breeds, has unique characteristics, especially disease resistance and economic traits, but the epigenetic basis underlying zebu’s merit traits is still to be studied. Therefore, in this study, we used Yunnan zebu as a representative of zebu and analyzed the differences in DNA methylation between zebu and taurine cattle (*Bos taurus*) using whole-genome bisulfite sequencing technology. We found no significant differences in the DNA methylation patterns between the two cattle types but identified many pathways and candidate genes associated with disease, disease resistance, and economic traits. The results of this study may provide a foundation for further screening of characteristic epigenetic molecular markers in zebu, thus providing theoretical support for the genetic improvement of domestic cattle.

**Abstract:**

Disease risk is a persistent problem in domestic cattle farming, while economic traits are the main concern. This study aimed to reveal the epigenetic basis for differences between zebu (*Bos indicus*) and taurine cattle (*Bos taurus*) in disease, disease resistance, and economic traits, and provide a theoretical basis for the genetic improvement of domestic cattle. In this study, whole genome bisulfite sequencing (WGBS) was used to analyze the whole-genome methylation of spleen and liver samples from Yunnan zebu and Holstein cattle. In the genome-wide methylation pattern analysis, it was found that the methylation pattern of all samples was dominated by the CG type, which accounted for >94.9%. The DNA methylation levels of different functional regions and transcriptional elements in the CG background varied widely. However, the methylation levels of different samples in the same functional regions or transcriptional elements did not differ significantly. In addition, we identified a large number of differentially methylation region (DMR) in both the spleen and liver groups, of which 4713 and 4663 were annotated to functional elements, and most of them were annotated to the intronic and exonic regions of genes. GO and KEGG functional analysis of the same differentially methylation region (DMG) in the spleen and liver groups revealed that significantly enriched pathways were involved in neurological, disease, and growth functions. As a result of the results of DMR localization, we screened six genes (*DNM3*, *INPP4B*, *PLD*, *PCYT1B*, *KCNN2*, and *SLIT3*) that were tissue-specific candidates for economic traits, disease, and disease resistance in Yunnan zebu. In this study, DNA methylation was used to construct links between genotypes and phenotypes in domestic cattle, providing useful information for further screening of epigenetic molecular markers in zebu and taurine cattle.

## 1. Introduction

DNA methylation was the earliest discovered epigenetic regulation mechanism [1] and is present in most eukaryotes [2]. DNA methylation is an epigenetic modification required for a variety of biological processes such as the regulation of gene expression, developmental regulation, gene imprinting, X chromosome inactivation, aging, and cell differentiation in eukaryotes [3]. The level of DNA methylation may vary somewhat between cells, tissues, or individuals, and even between different developmental periods of the same cell or individual, as shown by Ohgane et al., who demonstrated the presence of a large number of tissue-dependent differentially methylated regions (T-dependent) in unique sequences of the mammalian genome, shaping a unique DNA methylation profile for each tissue or cell type [4]. Meanwhile, DNA methylation plays an important role in complex mammalian representations [5], such as in the placenta, where site-specific CpG exhibits hypomethylation [6], and in skeletal muscle, where DNA methylation contributes to the suppression of gene expression in several biological processes and diseases [7]. In contrast, abnormal DNA methylation is commonly found in most tumors [8], such as breast cancer [9], prostate cancer [10], cervical cancer [11], bladder cancer [12], etc. In addition, abnormal DNA methylation may also contribute to neurodegenerative diseases [13], cardiovascular diseases [14], autoimmune diseases [15], and other diseases. In recent years, it has been demonstrated that DNA methylation also influences economic traits, disease resistance, and disease development in poultry and livestock [16,17]. For example, Wang et al. demonstrated that milk and protein yields were associated with genome-wide DNA methylation levels in lactating dairy cows [18]. Dong et al. proposed some potential candidate genes for milk production performance in dairy cows [19]. Kiefer et al. demonstrated, for the first time, a genome-wide association between DNA methylation and perinatal mortality in cattle [20]. One study also found a significant reduction in the methylation level of the promoter of miR-29b, a gene that interferes with the replication of the bovine viral diarrhea virus, in cell samples infected with the disease [21]. It has also been reported that there are potential regulatory roles for DNA methylation in cows infected with mastitis [22]. Nowadays, DNA methylation analysis is widely used in genetic studies of cattle; however, few studies have been conducted to resolve genome-wide DNA methylation differences between different subspecies of domestic cattle from this perspective.

Domestic cattle have contributed greatly to today’s agricultural and meat and dairy food industries, including the two species zebu (*Bos indicus*) and taurine cattle (*Bos taurus*) [23,24]. Although the two species share a common ancestor, they are still in an evolutionary stage, and the mechanism of reproductive isolation between the two has not been fully established. Selection pressures caused by complex evolutionary and domestication processes have led to significant differences in the phenotypes, economic traits, and disease resistance between the two species [25]. Zebu, also known as humped cattle, are widely distributed in tropical and subtropical climates, while nearly half of the world’s zebu are located in South Asia. Its biggest physical feature is that there is a hump-like bulge of muscle tissue between the neck and back, dewlaps from the throat to the abdomen, and large, pendulous ears [26]. As a result of by environmental selective pressures, zebu have better heat resistance, tolerance to mosquito attacks, and parasite resistance, and they are more resistant to roughage [27,28]. Raffaele et al. reported that zebu have better anti-tick and anti-tick-borne microorganism (TBM) ability than taurine cattle, which is related to the regulation of the host’s immune system [29]. However, zebu grow more slowly and are more prone to trypanosomiasis than taurine cattle [30,31]. It was also tested, and the results showed that taurine deep pectoral and semitendinosus muscles cooked as roasts had a lower WBSF than those of zebu; that is, the tenderness of zebu is lower [32], and zebu carcasses are lighter [33].

Yunnan is a frontier region bordering South and Southeast Asia and a mixed zone of taurine cattle and zebu. Yunnan zebu, also known as Yunnan humped cattle, are distributed only in Yunnan Province; they are the only zebu subspecies in China and are one of the most precious breed resources in China. The Yunnan zebu is a relatively pure zebu subspecies [34], and Li et al. also confirmed that the level of genomic diversity in Yunnan zebu cattle is not affected by taurine cattle [35]. Thus, the Yunnan zebu has almost all the characteristics of zebu, making it an ideal population for studying zebu’s genetic traits in China. However, the current studies on Yunnan zebu are only limited to the whole genome, transcriptome, proteome, and genetic evolution, and little is known about whether there are epigenetic differences between Yunnan zebu and taurine cattle, especially differences in DNA methylation.

Therefore, we used whole-genome bisulfite sequencing (WGBS) to perform a comparative DNA methylation analysis of the liver and spleen tissues between Yunnan zebu and Holstein cattle. The purpose was to characterize the DNA methylation profiles and differential methylation regions between two types of cattle under tissue-specific conditions and identify tissue-specific candidate genes related to economic traits, disease, and disease resistance. The results of this study may lay the foundation for further excavation of the epigenetic molecular markers in zebu and provide a theoretical basis for the genetic improvement of domestic cattle.

## 2. Materials and Methods

### 2.1. Ethical Statement

All liver and spleen samples collected from experimental animals in this study were approved by the Animal Research Ethics Committee of Yunnan University (approval number: ynucae20200316), and the experiments in this study were conducted according to the regulations and guidelines established by this committee.

### 2.2. Collection of Liver and Spleen Samples

Spleen samples (3 biological replicates) and liver samples (2 biological replicates) from 3 Yunnan zebu and 3 Holstein cattle were selected for WGBS and high-throughput sequencing. Yunnan zebu was used as the experimental group and Holstein cattle were used as the control group, both of which were harvested from Anning City, Kunming, Yunnan Province. Yunnan zebu weighed 214–263 kg and Holstein cattle weighed 549–607 kg. All study subjects were 4–5 years old, and were unrelated females in good health. Both cattle species were kept under the same feeding conditions for 2 months before sample collection, and the collection was completed within 10 min after the animals died. The samples were frozen and stored in liquid nitrogen immediately after collection and then transferred to a −80 °C refrigerator for backup, as shown in Table 1.

### 2.3. DNA Sample Preparation, Library Building, and WGBS Sequencing

DNA was extracted from 6 spleen samples and 4 liver samples using the Blood and Tissue Kit (QIAGEN). The genomic DNA was randomly broken into 200 bp sized DNA fragments using ultrasound and then treated by heavy sulfite conversion, where the unmethylated C in the DNA fragment was transformed to U. After transformation, the complete DNA methylation library was constructed by screening the DNA fragments and amplification by PCR techniques. Among them, the conversion rate of the spleen group was 99.54%; the conversion rate of the liver group was 99.55%. The PCR conditions were 95 °C for 3 min, followed by 35 cycles at 95 °C for 30 s each, annealing at 57 °C for 30 s and 72 °C for 30 s, and a final elongation step at 72 °C for 1 min. The primers were (forward primer) AATGATACGGCGACCACCGAGATCTACAC and (reverse primer) ATCTCGTATGCCGTCTTCTGCTTG. Finally, the constructed DNA libraries were quality-controlled and, after passing, sequenced by the Illumina HiSeq platform, with a sequencing strategy of PE150 and a sequencing depth of 30×.

### 2.4. Data Processing

Raw image data files were first converted to raw reads using CASAVA v.1.6 software, and then the reads containing conjoint sequences, low-quality sequences, and nitrogen (N) ratios of >10% were strictly filtered out to obtain clean reads that could be used for further bioinformatic analysis. Clean reads were compared with the reference genome using Bismark [36] v.0.19.0 software. The reference genome in this study was the *Bos indicus* genome (accession number: 3418) downloaded from the NCBI database (https://www.ncbi.nlm.nih.gov/genome/?term=Bos+indicus, accessed on 5 November 2018). The Bismark methylation extractor software in Bismark was used to extract the methylation information of the measured genomic DNA. Next, under the conditions of q < 0.00005 and a difference in the methylation between groups being greater than 0.1, DSS [37] software was used to identify the differentially methylated sites. DMRs were screened using eDMR [38] v.0.5.1 at *p* < 0.00005 and CpG number ≥ 3. The structure of the DMRs was annotated using the annotation information of the zebu genome to obtain the DMR-associated genes, Cross-tabulation analysis of DMG in the spleen and liver groups. Gene Ontology (GO) and Kyoto Encyclopedia of Genes and Genomes (KEGG) enrichment analysis (*p* ≤ 0.05) of these genes were based on DAVID (https://david.ncifcrf.gov/).

## 3. Results

### 3.1. WGBS and High-Throughput Sequencing Statistics

WGBS and high-throughput sequencing statistics were used to sequence 10 samples. Yunnan zebu and Holstein cattle had an average of 644,143,425 and 657,630,321 raw reads from their spleens, respectively, and Yunnan zebu and Holstein cattle had an average of 927,227,876 and 1,051,211,391 raw reads from their livers, respectively. After data quality control, 615,464,002, 618,558,002, 610,663,097, and 593,626,972 clean reads were obtained in turn. Using the *Bos indicus* genome as the reference genome, the WGBS sequencing results of 10 samples were compared and analyzed using Bismark v.0.19.0 software. The clean reads were then matched to the reference genome, and the results showed that the unique matching rate of the spleen group was over 60% in all cases, while the unique matching rate of the liver group was over 70% in all cases (Appendix A). We found that although the liver group detected more average raw reads than the spleen group, the liver group had a lower recognition rate of clean reads, and its unique mapped rate was higher than that of the spleen group.

### 3.2. Genome-Wide DNA Methylation Profiles in Bos indicus and Bos taurus

#### 3.2.1. Genome-Wide DNA Methylation Patterns

In the genome, there are three main patterns of DNA methylation at CpG sites (CG, CHG, and CHH), and the number and composition ratio of these three patterns reflect the characteristics of genome-wide methylation in a given species. The proportions of the effective C bases in this study were all over 50%, and the coverage was high. Further extraction of the DNA methylation information showed that there were 14,481,064 methylated cytosines in each sample on average. The main methylation pattern in all samples was CG, accounting for more than 94.95% of all methylated C; non-mCG (including CHG and CHH) modes existed, although these were few (Appendix A). The distribution of methylation levels in the table showed that domestic cattle have the highest overall CG background methylation levels, while CHG and CHH background methylation levels were extremely low or even close to zero.

To gain insight into the DNA methylation of domestic cattle, we depicted the genome-wide DNA methylation levels using the Cricos map (Figure 1, Appendix A). As seen in these plots, DNA methylation was present in almost the entire genome of the liver and spleen tissues, and the CG methylation levels were clearly distinct from the CHG and CHH methylation levels in the entire genome in each sample, with CG methylation patterns predominating. Overall, the CG methylation pattern was the predominant type of DNA methylation, and this finding was not influenced by the breed or tissue of the cattle.

#### 3.2.2. Methylation Levels in Different Functional Regions

The analysis showed that the CG pattern of methylation accounted for most of the overall methylation. To clearly understand the distribution trend of DNA methylation in the CG context of genomic regions, we counted the DNA methylation levels in six functional regions and seven transcription elements. The results showed that the methylation levels of different functional regions and transcription elements were significantly different. Among the functional regions, the 5’UTR region had the lowest methylation level, whereas the 3’UTR and gene regions had the highest average methylation level. Among the transcription elements, the first exon had the lowest methylation level, and the regions of the internal exon, internal intron, and last exon had the highest methylation levels. However, the differences in methylation levels among different samples in the same functional regions or transcriptional elements were not significant.

The DNA methylation levels of all samples in the six functional regions showed overall consistency (Figure 2a). The DNA methylation level in the 5’UTR region was the lowest, with an average of 36.28%. The methylation levels in the regions of the intron, 3’UTR, gene, intergenic, and exon showed little difference, with an average of 64.70%, 67.51%, 65.83%, 58.59%, and 65.33%, respectively, which means that the 3’UTR and gene regions had the highest average methylation levels. There were some differences in the methylation levels among samples from different regions, among which the difference between the intergenic region and the 3’UTR region was relatively large. A two-tailed Student’s *t*-test was used to compare the significant differences between subspecies and between tissues, including four comparison groups, namely, between Yunnan zebu’s livers and spleens, between Holstein cattle’s livers and spleens, between the livers of Yunnan zebu and Holstein cattle, and between the spleens of Yunnan zebu and Holstein cattle. The results showed that in the six functional areas, the significant *p*-values of the four groups ranged from 0.066 to 0.999, and the *p*-values were greater than 0.05, that is, the differences were not significant. Next, all the coding gene sequences were divided into seven different transcription element regions, the regions 2K upstream and downstream were equally divided into 40 copies, and the other transcription element regions were divided into 20 copies. We analyzed the DNA methylation level of the seven transcription elements in the CG background, including upstream, the first exon, the first intron, the internal exon, the internal intron, the last exon, and downstream. The results showed that (Figure 2b,c) the DNA methylation level near the transcription start site (TSS) was low, and a peak formed at the transcription start site (TSS). The lowest methylation level was maintained in the region of the first exon, and the DNA methylation level rose to about 60% from the first exon to the first intron. After that, the DNA methylation level in the internal exon, internal intron, last exon, and downstream regions remained stable at a high level of 60–80%, with no significant change. The results above indicate that although there were some differences in methylation levels among samples from different regions, the methylation levels in various functional regions and different transcription elements of the genomes of Yunnan zebu and Holstein cattle were relatively stable and less affected by tissue specificity and differences in the variety.

#### 3.2.3. Differential Methylation Regions (DMR)

DSS software was used to identify Differentially methylation cytosin (DMC) in two comparison groups, the spleen and liver of Yunnan zebu and Holstein cattle. In this case, the DSS was tested based on the β-binomial distribution model using the Wald test. Under the conditions of q < 0.00005 and a methylation difference between groups greater than 0.1, 294,212 DMC and 338,895 DMC were detected in the comparison between the spleens and livers of Yunnan zebu and Holstein cattle. The determined DMC was further screened. In total, 9474 DMRs were screened in the spleen group with *p* < 0.00005 and the number of CpG ≥3, and 12,329 DMRs were screened in the liver group (Table 2). In this study, the DMRs identified above were annotated using the annotation information of functional elements in the *Bos indicus* genome. Here, 2789 and 2768 differential methylation-related genes were annotated, respectively, in the spleen group and liver group, and 4713 and 4663 DMRs were annotated, respectively, on functional elements. Among them, in the spleen group, the methylation level of 1985 DMRs of Yunnan zebu was lower than that of Holstein cattle (hypo), and the methylation level of 2728 DMRs of Yunnan zebu was higher than that of Holstein cattle (hyper). In the liver group, there were 1895 hypo and 2768 hyper DMRs.

In this study, most DMRs were found in the intron regions (spleen group: 3677; liver group: 4300), followed by the exon regions (spleen group: 440; liver group: 216), and DMRs in other regions were relatively low (spleen group: 596; liver group: 147). Other regions of the spleen included the CGI, promoter, 3’UTR, and 5’UTR, and other regions of the liver referred to as upstream (Figure 3a). There was no significant difference in the number of DMRs annotated on the functional elements of the *Bos indicus* genome between the two comparison groups, but the function of the relevant tissue-specific DMG needs further analysis.

### 3.3. Functional Analysis of Differentially Methylation Genes

We strictly screened the DMR. When the number of CpGs was ≥ 5, 1702 DMRs were screened in the spleen group and 2111 DMRs were screened in the liver group. Under these conditions, 1164 DMGs were screened in the spleen group, and 1460 DMGs were screened in the liver group. Venn mapping of these genes was performed (Figure 3b). It was found that there were 344 identical DMGs in the two groups. GO enrichment and KEGG pathway analyses were performed on them, and 65 GO entries (Appendix A) were significantly enriched (*p* < 0.05). We mapped the top 20 items with extremely significant enrichment (Figure 3c), and 20 KEGG pathways (Figure 3d, Appendix A) were significantly enriched (*p* < 0.05). The results of the GO analysis include 26 biological processes (BP), mainly focusing on single-organism cellular processes, cellular component organization, and cellular responses to a stimulus. Thirty cellular components (CC) were found, mainly around the intracellular part, plasma membrane, and neuron part. The nine molecular functions (MF) mainly focused on protein binding, small molecule binding, and carbohydrate derivative binding. The KEGG enrichment pathway mainly includes: axon guidance (bta04360), the phospholipase D signaling pathway (bta04072), insulin secretion (bta04911), choline metabolism in cancer (bta05231), etc.

Through the combination of genome-wide DMR mapping and function analysis of the DMG, it was found that most of the genes identified as enriched by the GO and KEGG analyses were located in intron regions, and a few were located in the promoter, 5’UTR, and exon regions, such as *DNM3*, *INPP4B*, *KCNN2*, *PCYT1B*, *PLD*, and *SLIT3*. In the spleen of Yunnan zebu, the methylation levels in the promoter region of *DNM3* and the 5’UTR region and exon region of *PCYT1B* were lower than those in the spleen group of Holstein cattle, and the methylation levels in the exon region of *KCNN2* and *SLIT3* were higher than those in the spleen group of Holstein cattle. However, only the introns of the above genes were annotated with DMR in the liver group. In Yunnan zebu livers, the methylation levels of the exon region of *INPP4B* and *PLD1* were lower than thos of the Holstein liver group, while DMR were only annotated at the intron of these genes in the spleen group.

## 4. Discussion

In recent years, a few studies have addressed DNA methylation in zebu versus taurine cattle. For example, Sana et al. found that the stress gene expression of zebu was relatively high, based on research into Indian native zebu cattle and crossbred cattle, and DNA methylation may play a role in regulating the expression of certain genes involved in the stress response pathway [39]. It was also reported that beef tenderness in zebu may be influenced by the level of DNA methylation [40]. Undoubtedly, all the above studies provide useful information for epigenetic research on domestic cattle, but the findings of this research are not comprehensive. There are still some differences between the two domestic cattle subspecies, zebu and *Bos indicus*, in various aspects, such as production performance, phenotype, and disease resistance [41]. Therefore, given the current phenomenon of insufficient studies on differences in DNA methylation between the two cattle, we used the WGBS technique to analyze the existence of differences methylation and reveal the relationship between their epigenetic and phenotypic characteristics.

In the spleen and liver tissues of Yunnan zebu and Holstein cattle, the methylation levels of the CG type accounted for the largest proportion, while those of the CHH and CHG types were the smallest, close to 0. A large number of studies have also confirmed that the DNA methylation pattern of mammals is dominated by the CG type [42], including humans [43,44], mice [45], and pigs [46]. In the CG context, the methylation level of functional regions is significantly different. CG methylation at different positions of the genome also has different functions [47]. The methylation level of the 5’UTR was the lowest, and the methylation levels of the 3’UTR and gene regions were both high. However, the methylation level of the 3’UTR region is often related to gene expression. For example, Michael et al. found a strong positive correlation between 3’UTR DNA methylation of specific genes and increased gene expression [48]. Among the transcription elements, the methylation levels in the regions of the internal exon, internal intron, and last exon were the highest overall. Methylation patterns similar to the above are also very common in other mammals [49,50,51,52]. There was no significant difference in the methylation levels among different samples in the same region, and the DNA methylation levels in six functional regions and seven transcription elements in all samples showed overall consistency. It is speculated that the distribution trend of DNA methylation levels in the genome of domestic cattle is less affected by the breed and tissue-specificity, but this conclusion needs to be confirmed by detecting DNA methylation in more tissues from more subspecies of cattle. Although the number of differential methylation regions screened from the spleen group and liver group differed greatly, the two groups of DMR were mainly annotated as intron and exon regions, respectively. Studies have shown that the DNA methylation level of exons and introns is relatively stable, showing a weak positive and negative correlation with gene expression [53]. Introns, in particular, were once considered uncertain sequences and as genomic junk. However, more and more researchers have gradually realized that introns have important biological functions [54], which are closely related to gene expression. However, due to the large scale of differential methylation regions in the introns, it is difficult to screen functional differential methylation genes.

The same DMG identified in the liver group and spleen group were found to be significantly enriched in the GO and KEGG analyses, and were mainly related to the nervous system, disease, growth, and other functions. According to the results of the functional analysis and DMR mapping, most of the genes enriched in the GO and KEGG analyses were located in the intron regions, and a few were located in the promoter, 5’UTR, and exon regions. In this study, six tissue-specific candidate genes were screened, including *DNM3*, *INPP4B*, *PLD*, *PCYT1B*, *KCNN2*, and *SLIT3*. *DNM3* plays an inhibitory role in various malignant tumors in humans. In particular, the promoter of *DNM3* is hypermethylated in hepatocellular carcinoma [55,56]. DNA methylation in the promoter region is considered to inhibit gene expression [47]. In addition, *DNM3* is also upregulated in T-cell lymphoma (Sézary syndrome) originating from the skin [57]. The results of this study showed that the methylation level in the promoter region of the *DNM3* gene in the spleens of Yunnan zebu was lower than that in the spleens of Holstein cattle, while *DNM3* was not studied in domestic cattle. Therefore, we hypothesized that hypermethylation of the DNM3 promoter region may lead to high expression of DNM3 and thus affect tumorigenesis and development in zebu. Chinju et al. found that the expression of *INPP4B* in Angus cattle with a high residual feed intake (RFI) was lower than that in low-RFI cattle [58]. Our research results showed that the methylation level of the exon region of *INPP4B* in the livers of Yunnan zebu cattle was lower than that of Holstein cattle, and the research of other scientists has shown that RFI is not different among cattle subspecies [59,60]. Therefore, is the expression of *INPP4B* related to its methylation level, thus affecting the RFI of cattle? Further research on this gene may help to improve feed efficiency and reduce animal production costs. The expression of *PLD* is an influential factor in the inflammatory response [61,62]. Previous studies have shown that there are differences in the inflammatory response between zebu and taurine cattle. For example, the parts of ordinary cattle infected with ticks will have a greater cellular inflammatory response [63,64]. However, in the liver of Yunnan zebu, the methylation level of the exon region of *PLD1* is low, so it is inferred that the low methylation of *PLD* may be a factor in the difference in the inflammatory response between the two kinds of cattle. *PCYT1B* is related to the pH of chicken [65], and the decrease in pH after the death of animals is a key factor affecting meat quality [66]. According to a study on the change in pH after the death of Brahman cattle and Angus cattle, Brahman cattle showed a higher pH in the longissimus lumborum (LL) 3 h after death [67]. In the spleens of Yunnan zebu in this study, the methylation level in the 5’UTR region and exon region of *PCYT1B* was low. indicating that the low methylation of *PCYT1B* may affect the rate of decline rate in the postmortem pH value of zebu, further affecting the meat quality of zebu. The variation in *KCNN2* is related to movement disorders in rodents and humans [68]. The hypermethylation level of the exon region of *KCNN2* in the spleen of Yunnan zebu may inhibit or enhance the expression of *KCNN2*, thus affecting the movement of zebu. Laure et al. found that there is a differential methylation gene, *SLIT3*, in small ruminant sheep [69], and this gene is related to muscle development [70]. Therefore, we found that the higher methylation level of the exon region of *DNM3* in Yunnan zebu spleens may be related to beef development. At the same time, the genes *DNM3*, *KCNN2*, *SLIT3*, and *PCYT1B* only had DMR at the introns in the liver group, and *INPP4B* and *PLD1* only had DMR at the introns in the spleen group. That is, the functional regions of these six DMGs were different between the spleen group and the liver group; this indicates that there are tissue-dependent DMR and DMG in the cattle genomes in this study. In conclusion, the candidate genes we screened provide clues for the epigenetic and molecular target analysis of zebu in the future. Further functional experiments are needed to explore the specific functions of these genes and their roles in the economic, disease, and disease resistance traits of zebu.

## 5. Conclusions

In conclusion, through WGBS and high-throughput sequencing, this study showed that the methylation patterns of all domestic cattle samples were dominated by the CG type. The DNA methylation profiles were less affected by the cattle subspecies and tissue, and the differential methylation regions were mainly located in the introns and exons. At the same time, the GO and KEGG functional analyses showed that the DMG were mainly enriched in the nervous system, disease, and growth-related pathways. Through a combination of analyses of the differential methylation regions and functional analysis, we obtained six tissue-specific candidate genes (*DNM3*, *INPP4B*, *PLD*, *PCYT1B*, *KCNN2*, and *SLIT3*) related to economic, disease, and disease resistance traits in Yunnan zebu. This may lay the foundation for epigenetic research and related molecular marker mining of the DNA methylation levels of zebu.

## Figures and Tables

**Figure 1 animals-13-00203-f001:**
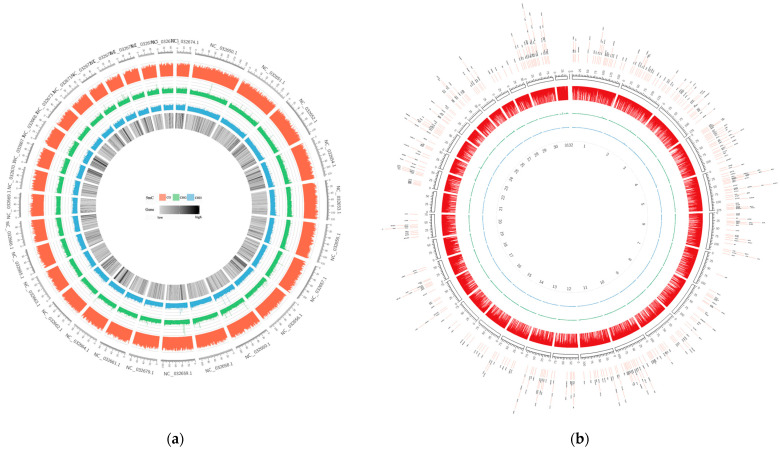
Whole-genome DNA methylation patterns of Yunnan zebu spleens and livers from No. 33. The red circle represents the CG methylation level, the green circle represents the CHG methylation level, and the blue circle represents the CHH methylation level. (**a**) PBI33. The outermost circle in the figure represents the chromosome scale, and the innermost circle represents the gene density. (**b**) GBI33. The outermost DMR in the figure are the chromosome scales, and the innermost numbers represent the chromosome numbers.

**Figure 2 animals-13-00203-f002:**
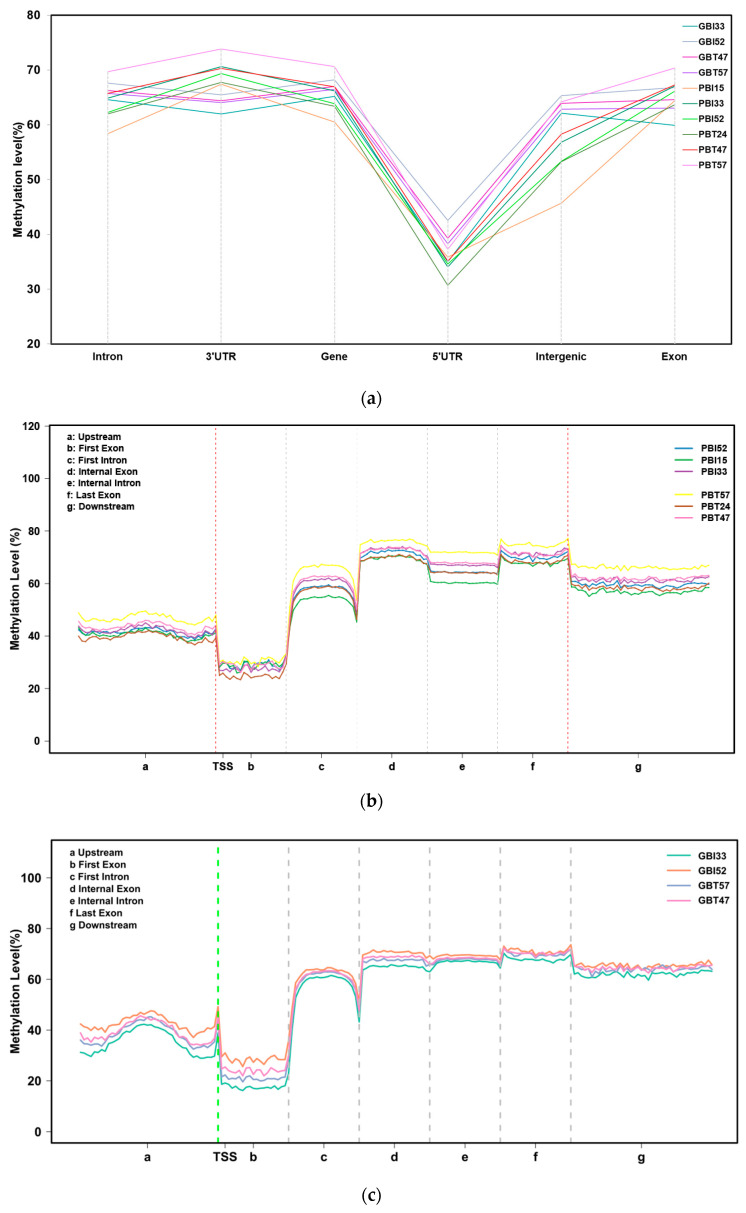
Methylation levels in different functional areas. (**a**) DNA methylation levels of 10 samples in different functional regions of the genome under a CG background. The x-axis indicates the names of the functional regions. The y-axis indicates the methylation level (%). (**b**) Distribution of DNA methylation levels on different transcription elements in the CpG background of the spleen group. The x-axis indicates the different regions of gene transcription elements; the y-axis indicates the methylation level. (**c**) Distribution of DNA methylation levels on different transcription elements in the CpG background of the liver group.

**Figure 3 animals-13-00203-f003:**
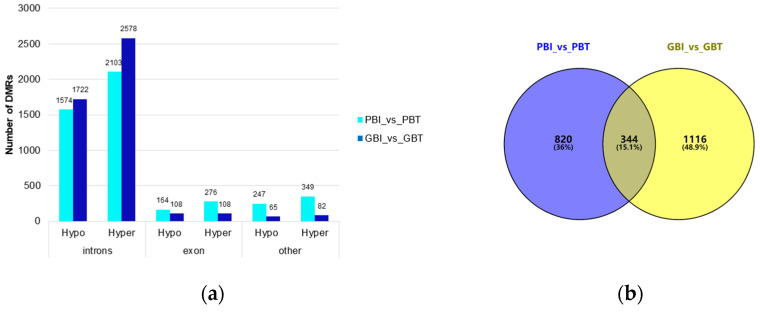
Genome-wide differential methylation regions and functional annotation of the differential methylation genes. (**a**) Distribution of DMR annotated to functional areas. The x-axis represents the number of DMR annotated to the corresponding functional regions, and the y-axis represents the functional regions of the genome. (**b**) Venn diagram of DMGs in the spleen group and liver group. PBI vs. PBT represents the spleen group of Yunnan zebu and Holstein cattle, and GBI vs. GBT represents the liver group of Yunnan zebu and Holstein cattle. (**c**) GO enrichment results of common differential methylation genes in the spleen and liver groups. The x-axis represents enriched factors, and the y-axis represents the name of the GO pathway. (**d**) KEGG enrichment results of common differential methylation genes in the spleen and liver groups. The x-axis represents enriched factors, and the y-axis represents the name of the KEGG pathway.

**Table 1 animals-13-00203-t001:** Details of the samples.

Samples	Latin Name of Species	Subspecies of Cattle	No.
Spleen	*Bos indicus*	Yunnan zebu	PBI15
	*Bos indicus*	Yunnan zebu	PBI33
	*Bos indicus*	Yunnan zebu	PBI52
	*Bos taurus*	Holstein cattle	PBT24
	*Bos taurus*	Holstein cattle	PBT47
	*Bos taurus*	Holstein cattle	PBT57
Liver	*Bos indicus*	Yunnan zebu	GBI33
	*Bos indicus*	Yunnan zebu	GBI52
	*Bos taurus*	Holstein cattle	GBT47
	*Bos taurus*	Holstein cattle	GBT57

**Table 2 animals-13-00203-t002:** Differentially methylated cytosine (DMC) and differential methylated regions (DMR) in the two comparisons.

Comparison	Number	DMC	DMR	Hypo-DMR	Hyper-DMR
Yunnan zebu spleens and Holstein cattle spleens	PBI vs. PBT	294,212	9474	4466	5008
Yunnan zebu livers and Holstein cattle livers	GBI vs. GBT	338,895	12,329	4754	7575

## Data Availability

The data and results generated from this project are fully available upon request.

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
