# Peer review of "Genome-Wide DNA Methylation Differences between *Bos indicus* and *Bos taurus"

_animals, 2023, doi:10.3390/ani13020203_

Round 1

Reviewer 1 Report

Generally, the article is well-writen. Its strong point is, obiouvsly, the experimental protocol, including here the method of working which led to valuable results. Some deficiencies in English writing can be noted, for example some sentencies are too long and difficult to be understood (row 19 in abstract, row 46 - „The level of DNA methylation...” At row 70 „however” is noted with upper case letter altough the sign before is „;”. At row 211, „student” is noted with lower case letter, although is considered the „name” of a statistical test. A question can be related to the no of samples, and I wonder if the obtained results are not influenced by this small number. 

Author Response

Comments and Suggestions for Authors:

Generally, the article is well-written. Its strong point is, obviously, the experimental protocol, including here the method of working which led to valuable results. Some deficiencies in English writing can be noted, for example, some sentences are too long and difficult to be understood row 19 in the abstract, row 46 - “The level of DNA methylation...” On row 70 “however” is noted with an upper case letter although the sign before is “; ”. In row 211, “student” is noted with lowercase letters, although is considered the “name” of a statistical test. A question can be related to the no of samples, and I wonder if the obtained results are not influenced by this small number.

General response: First of all, I would like to thank reviewer 1 for her/his valuable comments on this paper. Accordingly, we have carefully adjusted and revised the paper. Detailed responses as point-to-point as below:

1. Some deficiencies in English writing can be noted, for example, some sentences are too long and difficult to be understood row 19 in the abstract, row 46 - “The level of DNA methylation...” .

Response: Thank you for your valuable suggestions. We found many shortcomings in the English writing, so we have made detailed changes to this manuscript. This includes a revision of the original row 19, row 46- “The level of DNA methylation...” was revised and restructured. In addition, we revised and corrected other parts of the English writing, including changes and sentence breaks on rows 27, 44, 54, 250, 277, 395, 473, 480, and 481, as well as multiple other grammatical changes.

2. At row 70 “however” is noted with upper case letter although the sign before is “ ; ”.

Response: We have corrected "however" after “ ; ” in row 70 and corrected the same problem in the article.

3. At row 211, “student” is noted with lower case letter, although is considered the “name” of a statistical test.

Response: We have changed “student's t-test” to “Student's t-test” in row 211.

4. A question can be related to the no. of samples, and I wonder if the obtained results are not influenced by this small number.

Response: You may have considered that 2 experimental replicates for differential methylation analysis may only differ between individuals and not reflect the population characteristics of the two subspecies. We explain the corresponding results here. In the present study, our findings of 2 experimental replicates of the dry viscera group and 3 experimental replicates of the spleen group yielded consistent results, mainly in the following results:

(1) Although there was a difference in the number of raw reads and clean reads between the spleen and liver groups, there was no significant difference in the DNA methylation pattern and DNA methylation profiles between the two groups. The methylation patterns of both were dominated by the CG type. the DNA methylation levels of different functional regions and transcriptional elements in the CG background were both more different. And both showed the lowest methylation levels in the 5'UTR region and the highest average methylation levels in the 3'UTR and Gene regions in the functional regions. Among transcription elements, the methylation level of the First exon was the lowest, while that of the Internal exon, Internal intron, and Last exon regions was the highest.

(2) The difference in the number of DMR and DMG detected was not significant, especially the number of DMR (spleen group: 1702; liver group: 2111) and DMG (spleen group: 1164; liver group: 1460) after performing rigorous screening.

(3) We performed gene intersection of the above DMGs and found 344 common genes, mainly associated with neurological, disease, and growth functions.

In addition, the NIH National Institutes of Health Roadmap Epigenomics Project recommends using 2 biological replicates for DNA methylation mapping studies of cells and tissues of higher organisms and setting the sequencing depth to 30× (http://www.roadmap pigenomics.org/ protocols). and the effect of the WGBS method on experiments using 1, 2, or 3 biological replicates has been validated in an article (https://www.webofscience.com/wos/alldb/full-record/WOS:000350670300023). It was found that when the sequencing depth reached 10× or more, the biological repeats were elevated from 1 to 3, and the correct rate, error incidence, and DMR detection rate of the sequences showed a slowing trend, and the detection sensitivity of the correct rate was similar when the biological repeats were 2 and 3. It can be concluded that biological replicates 2 and 3 in this experiment did not have a significant effect on the assay results. Therefore, our other idea is also to demonstrate the effect of biological replicates 2 and 3 on the experimental results of this study.

Again, thank you for your thoughtful comments and suggestions.

Reviewer 2 Report

Comments: The manuscript entitled “Genome-wide DNA methylation differences between Bos indicus and Bos taurus by Xiaona Chen et al. has analyze the whole genome methylation of spleen and liver samples from Yunnan Zebu and Holstein cattle by WGBS. The design of this study is reasonable. The major concerns are as follows:

1.     Why did the experimental animals have 2 biological replicates for liver samples and 3 biological replicates for spleen samples in this study?

2.     Are there any other details about the experimental animals besides their age and sex provided by the authors? Such as body weight, health status, etc.

3.     Can you please explain why blood is not used for genomic DNA extraction but liver and spleen tissues? Is there a reason for this?

4.     Line 132-133, what were the DNA quality scores.

5.     In 2.4. Data processing. Details about the functional analysis, including the software used for the enrichment analysis, the conditions, and the p-value should be mentioned and not just appear in the results. In addition, GO and KEGG analyses should be written in full when they first appear in the manuscript.

6.     Line 214-216, What four groups are included in the “differences of four groups were not significant”

7.     Line 243-245, The identification conditions of DMC should be mentioned in the Materials and Methods.

8.     Similarly, in lines 268-269, the screening conditions for DMRs should be mentioned in the Materials and Methods.

9.     The quality of Fig 3 is not high enough, please provide a higher quality figure.

10.  Can the results of KEGG analysis be visualized with bubble plots?

11. Line 403, gene should be italicized.

Author Response

Reviewer 2’s Comments and Suggestions for Authors:

Comments: The manuscript entitled “Genome-wide DNA methylation differences between Bos indicus and Bos taurus” by Xiaona Chen et al. has analyzed the whole genome methylation of spleen and liver samples from Yunnan Zebu and Holstein cattle by WGBS. The design of this study is reasonable. The major concerns are as follows:

  1. Why did the experimental animals have 2 biological replicates for liver samples and 3 biological replicates for spleen samples in this study?
  2. Are there any other details about the experimental animals besides their age and sex provided by the authors? Such as body weight, health status, etc.
  3. Can you please explain why blood is not used for genomic DNA extraction but for liver and spleen tissues? Is there a reason for this?
  4. Line 132-133, what were the DNA quality scores.
  5. In 2.4. Data processing. Details about the functional analysis, including the software used for the enrichment analysis, the conditions, and the p-value should be mentioned and not just appear in the results. In addition, GO and KEGG analyses should be written in full when they first appear in the manuscript.
  6. Line 214-216, What four groups are included in the “differences of four groups were not significant”
  7. Line 243-245, The identification conditions of DMC should be mentioned in the Materials and Methods.
  8. Similarly, in lines 268-269, the screening conditions for DMRs should be mentioned in the Materials and Methods.
  9. The quality of Fig 3 is not high enough, please provide a higher-quality figure.
  10. Can the results of KEGG analysis be visualized with bubble plots?
  11. Line 403, gene names should be italicized.

General response: First of all, we would like to thank reviewer 2 for her/his valuable comments on this paper. Accordingly, we have carefully adjusted and revised the paper. Detailed responses as point-to-point as below:

1. Why did the experimental animals have 2 biological replicates for liver samples and 3 biological replicates for spleen samples in this study?

Response: You may have considered that differential methylation analysis using 2 experimental replicates may not be sufficient to reflect the population characteristics of the two subspecies. We have made the following explanations here:

The first objective reason was that during the quality check of DNA concentration and integrity of the collected pretreated samples, two samples in the liver group were found to be of low quality, including liver tissues from No. 15 Yunnan Zebu (GBI15) and No. 24 Holstein cattle (GBT24), which were not suitable for further study. The second is because the NIH National Institutes of Health Roadmap Epigenomics Project recommends using 2 biological replicates for DNA methylation mapping studies of cells and tissues of higher organisms and setting the sequencing depth to 30× (http://www.roadmap pigenomics.org/ protocols). and the effect of the WGBS method on experiments using 1, 2, or 3 biological replicates has been validated in an article (https://www.webofscience.com/wos/alldb/full-record/WOS:000350670300023). It was found that when the sequencing depth reached 10× or more, the biological repeats were elevated from 1 to 3, and the correct rate, error incidence, and DMR detection rate of the sequences showed a slowing trend, and the detection sensitivity of the correct rate was similar when the biological repeats were 2 and 3. It can be concluded that biological replicates 2 and 3 in this experiment did not have a significant effect on the assay results. Therefore, another idea we had, for objective reasons that are difficult to change, was also to demonstrate the effect of biological replicates 2 and 3 on the experimental results of this study.

2. Are there any other details about the experimental animals besides their age and sex provided by the authors? Such as body weight, health status, etc.

Response: We added details of the weight and health status of the study subjects to the manuscript. The details are: Yunnan Zebu weighed between 214-263 kg and Holstein cattle weighed between 559-607 kg. All study subjects were 4-5 years old, unrelated females in good health.

3. Can you please explain why blood is not used for genomic DNA extraction but for liver and spleen tissues? Is there a reason for this?

Response: Our study is based on the results of our previous research on Yunnan Zebu and other results in the literature. Because Zebu is more docile and hardier, easy to drive, and heat resistant. Most importantly, Zebu is highly resistant to ectoparasites such as ticks, mites, bull flies, and maggots, as well as to certain infectious and common diseases. On the other hand, the liver functions as the largest detoxification organ by storing glycogen, regulating the metabolism of proteins, lipids, and carbohydrates, and detoxifying poisons and waste products produced or ingested by the body. The spleen has a very rich blood circulation and is the largest lymphatic organ, so it is the most important immune organ in the body. Our initial aim was to investigate the epigenetic mechanisms associated with disease resistance in Yunnan Zebu through differential methylation analysis, so we selected the detoxification organ (liver) and the immune organ (spleen) for our experiments.

4. Line 132-133, what were the DNA quality scores?

Response: Thank you very much for your question, my understanding of this issue may not be thorough enough, so I have made the following reply for the time being, if there are any outstanding matters please contact us in time.

Your question about DNA quality scoring is probably to understand how the heavy sulfite treatment works, we add lambda DNA to every methylation build to evaluate the effect of the heavy sulfite treatment. Therefore, we provide here the value (Conversion Rate) used to determine the treatment of heavy sulfites. The Conversion Rate for the spleen group was 99.54%; for the liver group, it was 99.55%. Also, we have added this result to the manuscript.

5. In 2.4. Data processing. Details about the functional analysis, including the software used for the enrichment analysis, the conditions, and the p-value should be mentioned and not just appear in the results. In addition, GO and KEGG analyses should be written in full when they first appear in the manuscript.

Response: Based on your suggestions, we have added detailed information in 2.4. data processing, including software, URLs, conditions and p-values, and added full spellings where GO and KEGG first appear.

6. Line 214-216, What four groups are included in the “differences of four groups were not significant”

Response: The "differences of four groups were not significant" in the manuscript include: Yunnan Zebu liver and spleen, Holstein cattle liver and spleen, between the liver of Yunnan Zebu and Holstein cattle, between the spleen of Yunnan Zebu and Holstein cattle. In addition, the corresponding unclear expressions in the text have been revised.

7. Line 243-245, The identification conditions of DMC should be mentioned in the Materials and Methods.

Response: The identification conditions for DMC have been added to the Materials and Methods section based on your suggestions. The specific additions are as follows: Under the condition of q<0.00005, the methylation difference between groups was greater than 0.1.

8. Similarly, in lines 268-269, the screening conditions for DMRs should be mentioned in the Materials and Methods.

Response: The conditions for the identification of DMRs have been added to the Materials and Methods section based on your suggestions. The specific additions are as follows: DMRs were screened using eDMR v.0.5.1 at p<0.00005 and CpG number ≥3.

9. The quality of Fig 3 is not high enough, please provide a high-quality figure.

Response: Accordingly, we have reformatted Figure 3 and replaced the original GO bar chart in Figure 3c with a GO enrichment bubble chart (only the first 20 entries significantly enriched are plotted). In addition, we have increased the clarity of the four sub-figures in Figure 3.

10. Can the results of KEGG analysis be visualized with bubble plots?

Response: Based on your suggestion and to visualize the KEGG results, we have added Figure 3d KEGG bubble plots (20 KEGG pathways in total).

11. Line 403, gene names should be italicized.

Response: The gene names have been italicized based on your suggestion.

Again, thank you for your thoughtful comments and suggestions.

Reviewer 3 Report

1Why choose the spleen

2 Why the number of samples of spleen tissue and liver tissue is different

3. Cite relevant references for software and describe settings in line 242.

4. What does q<0.00005 mean? In line 243.

5. In the discussion section, the functions of DNM3, INPP4B, PLD, PCYT1B, KCNN2, and SLIT3 genes should be discussed

Author Response

Reviewer 3’s Comments and Suggestions for Authors:

  1. Why choose the spleen?
  2. Why the number of samples of spleen tissue and liver tissue is different?
  3. Cite relevant references for software and describe settings in line 242.
  4. What does q<0.00005 mean? In line 243.
  5. In the discussion section, the functions of DNM3, INPP4B, PLD, PCYT1B, KCNN2, and SLIT3 genes should be discussed

General response: First of all, we would like to thank reviewer 3 for her/his valuable comments on this paper. Accordingly, we have carefully adjusted and revised the paper. Detailed responses as point-to-point as below:

  1. Why choose the spleen?

Response: The content of our study is mainly based on the results of our previous research on Yunnan Zebu and other results in the literature. Because Zebu is more docile and hardier, easy to handle, heat-resistant, and most importantly, they have strong resistance to ectoparasites such as ticks, mites, bull flies and maggots, as well as strong resistance to certain infectious and common diseases. The spleen, on the other hand, has a very rich blood circulation and is the largest lymphatic organ, it is also the most important immune organ in the body. The initial aim of our study was to investigate the epigenetic mechanisms associated with disease resistance in tumor-bearing cattle through differential methylation analysis in Yunnan Zebu, so the immune organ (spleen) was selected for the experiment.

  1. Why the number of samples of spleen tissue and liver tissue is different?

Response: You may have considered that differential methylation analysis using 2 experimental replicates may not be sufficient to reflect the population characteristics of the two subspecies, or that different sample sizes may affect the reliability of the results. We have provided the following explanation here.

The first objective reason was that during the quality check of DNA concentration and integrity of the collected pretreated samples. Two samples in the liver group were found to be of low quality, including liver tissues from No. 15 Yunnan Zebu (GBI15) and No. 24 Holstein cattle (GBT24), which were not suitable for further study. The second is because the NIH National Institutes of Health Roadmap Epigenomics Project recommends using 2 biological replicates for DNA methylation mapping studies of cells and tissues of higher organisms and setting the sequencing depth to 30× (http://www.roadmap pigenomics.org/ protocols). and the effect of the WGBS method on experiments using 1, 2, or 3 biological replicates has been validated in an article (https://www.webofscience.com/wos/alldb/full-record/WOS:000350670300023). It was found that when the sequencing depth reached 10× or more, the biological repeats were elevated from 1 to 3, and the correct rate, error incidence and DMR detection rate of the sequences showed a slowing trend, and the detection sensitivity of the correct rate was similar when the biological repeats were 2 and 3. It can be concluded that biological replicates 2 and 3 in this experiment did not have a significant effect on the assay results. Therefore, another idea we had, for objective reasons that are difficult to change, was also to demonstrate the effect of biological replicates 2 and 3 on the experimental results through this study.

  1. Cite relevant references for software and describe settings in line 242.

Response: Based on your valuable comments, we have supplemented the Materials and Methods section with references to the software, including Bismark v0.16.3, DSS, and eDMR v.0.5.1. And the parameter settings of the DSS software in 242 lines are described.

  1. What does q<0.00005 mean? In line 243.

Response: The q-value (q-value) is the result of p-value correction. The q-value was used as a statistical test variable in this study because we derived too many significantly different data, and q < 0.00005 instead of q ≤ was also used to narrow the scope of the data analysis.

  1. In the discussion section, the functions of DNM3, INPP4B, PLD, PCYT1B, KCNN2, and SLIT3 genes should be discussed

Response: Thanks to your valuable suggestions, we have partially modified the discussion of six genes for the purpose of gene function discussion. including the possible involvement of DNM3 in the development of tumorigenesis in Zebu; INPP4B may affect the residual feed intake (RFI) in domestic cattle. PLD may be involved in the inflammatory response of Zebu; hypomethylation of PCYT1B may have affected the rate of postmortem pH decline in rumen cattle, thereby affecting the quality of rumen meat. KCNN2 may influence locomotion in domestic cattle; SLIT3 may be associated with muscle development in domestic cattle. Our previous description may have been too strong and we have revised it.

Again, thank you for your thoughtful comments and suggestions.

Reviewer 4 Report

The study compares the methylation rates between Bos indicus and B. taurus. The subject is interesting and of high importance. I have some minor comments and some recommendations that follow:  

Since zebu (Bos indicus) is a different species than Bos taurus, is it correct to characterise it as cattle breed? Maybe choose a different characterisation and apply this in the whole manuscript 

In 2.3 please provide some more details for the amplification using PCR techniques, such as primers, conditions etc

In line 139 please mention the accession number of the reference genome 

My basic criticism regarding the proposed conclusions is the speculations concerning the low methylation correlation with cancer. For instance I suggest to delete the lines 357-359. Similarly, the finding regarding the inflammatory response is too strong. Maybe also rephrase the lines 371-372

Author Response

Reviewer 4’s Comments and Suggestions for Authors:

The study compares the methylation rates between Bos indicus and B. taurus. The subject is interesting and of high importance. I have some minor comments and some recommendations that follow: 

Since zebu (Bos indicus) is a different species than Bos taurus, is it correct to characterize it as cattle breed? Maybe choose a different characterization and apply this in the whole manuscript.

In 2.3 please provide some more details for the amplification using PCR techniques, such as primers, conditions etc.

In line 139 please mention the accession number of the reference genome.

My basic criticism regarding the proposed conclusions is the speculations concerning the low methylation correlation with cancer. For instance, I suggest to delete the lines 357-359. Similarly, the finding regarding the inflammatory response is too strong. Maybe also rephrase the lines 371-372

General response: First of all, we would like to thank reviewer 2 for her/his valuable comments on this paper. Accordingly, we have carefully adjusted and revised the paper. Detailed responses as point-to-point as below:

  1. Since zebu (Bos indicus) is a different species than Bos taurus, is it correct to characterize it as cattle breed? Maybe choose a different characterization and apply this in the whole manuscript.

Response: Based on your valuable suggestions and descriptions in other literature, we have revised “breed” to “subspecies” in the manuscript.

  1. In 2.3 please provide some more details for the amplification using PCR techniques, such as primers, conditions etc.

Response: Relevant details have been added to the manuscript. PCR cycling conditions were as follows: 95°C for 3 min, followed by 35 cycles at 95°C for 30 s each; annealing at 57°C for 30 s, 72°C for 30 s, and a final elongation step at 72°C for 1 min. Primers (Forward primer: Forward primer: AATGATACGGCGACCACCGAGATCTACAC; Reverse primer: ATCTCGTATGCCGTCTTCTGCTTG).

  1. In line 139 please mention the accession number of the reference genome.

Response: Based on your comments, we have added the NCBI registration number of the Bos indicus reference genome to the manuscript. The accession number is 3418.

  1. My basic criticism regarding the proposed conclusions is the speculations concerning the low methylation correlation with cancer. For instance, I suggest to delete lines 357-359. Similarly, the finding regarding the inflammatory response is too strong. Maybe also rephrase lines 371-372.

Response: Based on your suggestion, we have deleted lines 357-359 and revised the corresponding section again. Also, we have revised the description of the "Inflammatory response findings" section, including lines 371-372.

Again, thank you for your thoughtful comments and suggestions.

Round 2

Reviewer 2 Report

The author has addressed all my concerns and I think the current manuscript can be used for publication in animals.